# CCL4 Functions as a Biomarker of Type 2 Airway Inflammation

**DOI:** 10.3390/biomedicines10081779

**Published:** 2022-07-23

**Authors:** Yoshiki Kobayashi, Hanh Hong Chu, Akira Kanda, Yasutaka Yun, Masami Shimono, Linh Manh Nguyen, Akitoshi Mitani, Kensuke Suzuki, Mikiya Asako, Hiroshi Iwai

**Affiliations:** 1Airway Disease Section, Department of Otorhinolaryngology, Kansai Medical University, Osaka 573-1010, Japan; chuh@hirakata.kmu.ac.jp (H.H.C.); akanda@hirakata.kmu.ac.jp (A.K.); yunys@hirakata.kmu.ac.jp (Y.Y.); takadama@kouri.kmu.ac.jp (M.S.); nguyenli@hirakata.kmu.ac.jp (L.M.N.); mitaniak@hirakata.kmu.ac.jp (A.M.); suzukken@hirakata.kmu.ac.jp (K.S.); asako@takii.kmu.ac.jp (M.A.); iwai@hirakata.kmu.ac.jp (H.I.); 2Allergy Center, Kansai Medical University Hospital, Hirakata, Osaka 573-1010, Japan

**Keywords:** bronchial asthma, CCL4, eosinophilic chronic rhinosinusitis, type 2 airway inflammation

## Abstract

Eosinophilic airway inflammatory disease is associated with bronchial asthma, with eosinophilic chronic rhinosinusitis (ECRS) typical of refractory type 2 airway inflammation. CCL4 produced at local inflammatory sites is involved in them via the accumulation and activation of type 2 inflammatory cells, including eosinophils. The detailed mechanism of CCL4 production remains unclear, and also the possibility it could function as a biomarker of type 2 airway inflammation remains unresolved. In this study, we evaluated CCL4 mRNA expression and production via the TSLP receptor (TSLPR) and toll-like receptors (TLRs) or proteinase-activated receptor-2 (PAR2) in BEAS-2B bronchial epithelial cells co-incubated with purified eosinophils or eosinophil peroxidase (EPX). We examined serum chemokine (CCL4, CCL11, CCL26, and CCL17) and total IgE serum levels, fractionated exhaled nitrogen oxide (FENO), and CCL4 expression in nasal polyps in patients with severe ECRS and asthma. CCL4 was induced by TSLP under eosinophilic inflammation. Furthermore, CCL4 was released via TLR3 signaling, which was enhanced by TSLP. CCL4 was mainly located in nasal polyp epithelial cells, while serum CCL4 levels were reduced after dupilumab treatment. Serum CCL4 levels were positively correlated with FENO, serum IgE, and CCL17 levels. Thus, CCL4 released from epithelial cells via the innate immune system during type 2 airway inflammation may function as a useful biomarker for the condition.

## 1. Introduction

Globally, bronchial asthma is a common disease. Approximately 5–10% of patients have severe asthma with inhaled corticosteroid (ICS)-resistant type 2 airway inflammation [1]. This inflammation is characterized by biomarkers such as eosinophils, IgE, and fractionated exhaled nitrogen oxide (FeNO) [2]. In Japan, 7–10% of patients with asthma who receive high dose ICS, long-acting β2 agonists, and additional treatments, including long-acting muscarinic agonists and/or leukotriene receptor antagonists, are defined as severe, with 80% of severe asthma consisting of type 2 inflammation [3].

Patients with severe asthma are associated with comorbidities which affect their asthma control. In particular, eosinophilic chronic rhinosinusitis (ECRS), a subtype of chronic rhinosinusitis with nasal polyps, which is also caused by type 2 immune responses, is closely associated with asthma [4]. Approximately 50% or more of severe asthma patients are associated with severe ECRS, whereas 80% or more of severe ECRS coexists with severe asthma [5,6,7,8]. Importantly, local corticosteroid sensitivity in patients with severe asthma and ECRS is markedly reduced, thus the conditions are difficult to control. As we previously indicated, co-incubation with eosinophils reduced corticosteroid sensitivity in bronchial epithelial cells; eosinophils in the airway induced corticosteroid resistance [9]. Therefore, it is important to elucidate mechanisms underpinning eosinophil accumulation in airways to inhibit eosinophil infiltration and restore corticosteroid sensitivity.

Several chemokines such as CCL3/MIP-1α, CCL4/MIP-1β, CCL5/RANTES and eotaxin family (CCL11/eotaxin1, CCL24/eotaxin2, and CCL26/eotaxin3) are involved in the pathogenesis of allergic asthma with type 2 immune responses [10]. Recently, we reported that CCL4, but not other chemokines, was abundantly released from activated eosinophils. Additionally, CCL4 activated eosinophils and acted as a chemokine against them [11,12]. Although CCL4 was mainly associated with mononuclear cells and released from them [11,13], activated eosinophils are also sources, especially in airway inflammation. Furthermore, we preliminarily confirmed that CCL4 was highly expressed in sinonasal epithelial cells from patients with ECRS [9].

In this study, we focused on CCL4 production mechanisms in airway epithelial cells, and examined the possibility of CCL4 being a biomarker for type 2 airway inflammation.

## 2. Materials and Methods

### 2.1. Cell Preparation

The human bronchial epithelial cell line BEAS-2B was obtained from the European Collection of Authenticated Cell Culture (Salisbury, UK). Eosinophils (purity > 98%) were isolated from the peripheral blood of healthy volunteers with mild eosinophilia (approximately 4–8% of total white blood cells) by negative selection using a MACS^©^ system with an Eosinophil Isolation Kit (Miltenyi Biotec, Bergisch Gladbach, Germany). BEAS-2B cells were co-incubated with purified eosinophils or recombinant eosinophil peroxidase (EPX, Proteintech, Rosemont, IL, USA) as required.

### 2.2. Quantitative RT-PCR

Total RNA extraction and reverse transcription were performed using a RNeasy Mini Kit (Qiagen, Hilden, Germany) and a PrimeScript RT MasterMix (Perfect Real Time; Takara Bio, Shiga, Japan). Transcript levels of *TSLP*, *IL-25*, *IL-33*, *CCL4*, *CCL5*, *CCL11*, *CCL26, TSLPR, TLR3, CCR5, ICAM-1,* and glyceraldehyde 3-phosphate dehydrogenase (*GAPDH*) were quantified by RT-PCR using a Rotor-Gene SYBR Green PCR kit (Qiagen, Hilden, Germany) on a Rotor-Gene Q HRM (Corbett Research, Cambridge, UK). Amplification primers (Eurofins Genomics, Tokyo, Japan) are shown (Appendix A).

### 2.3. Cell Lysis and Western Blotting

Cell protein extracts were prepared using modified RIPA buffer (50 mM Tris HCl pH 7.4, 1.0% NP-40, 0.25% Na-deoxycholate, 150 mM NaCl plus a complete protease inhibitor). Protein concentrations were determined using the BCA Protein Assay (Thermo Fisher Scientific, Rockford, IL, USA). Protein extracts were separated by SDS-PAGE (Bio-Rad, Hercules, CA, USA) and detected by western blot analysis using the Odyssey infrared imaging system (LI-COR Bioscience, Lincoln, NE, USA) according to the manufacturer’s instructions. The rabbit polyclonal antibody to CCR5 (GeneTex, Irvine, CA, USA) was used as a primary antibody. β-actin expression was used as a loading control.

### 2.4. Immunoassay for Chemokines and TSLP

CCL4 and TSLP levels in cell culture supernatants were measured using MIP-1β/CCL4 and TSLP ELISA kits (R&D Systems, Minneapolis, MN, USA). Serum CCL4/MIP-1β, CCL11/eotaxin1, CCL26/eotaxin3, and CCL17/TARC levels were evaluated using Milliplex kits (Merck, Darmstadt, Germany). Serum samples were obtained from patients with severe ECRS and asthma who received any biotherapy, including dupilumab.

### 2.5. Immunofluorescence Staining

After deparaffinization, rehydration and proteinase K-induced antigen retrieval, nasal polyp sections were blocked and stained with anti-EpCAM (Cell Signaling Technology, Danvers, MA, USA) and CCL4 (Bioss, Woburn, MA, USA), followed by incubation with Alexa 488 goat anti-rabbit and Alexa 647 goat anti-mouse antibodies (Jackson, PA, USA). After co-incubation overnight with purified eosinophils, BEAS-2B cells were fixed with 4% formaldehyde for 20 min, permeabilized, and blocked. Cells were then incubated with a mouse monoclonal antibody to CCR5 (Abcam, Cambridge, UK), followed by CF 647-labeled goat anti-mouse antibody incubation (Biotium, Fremont, CA, USA). Slides were visualized using a FV3000 confocal microscope (Olympus, Tokyo, Japan) with the identical settings. Control antibodies and Hoechst staining (Invitrogen, Paisley, UK) were included in each experiment. To evaluate the ratio intensity of CCL4 to EpCAM, ImageJ was used [14].

### 2.6. Statistical Analysis

Comparisons of two groups of data were performed using the Mann–Whitney U test or paired t-test. Correlation coefficients were calculated using the Spearman’s rank method. Other data were analyzed by one-way analysis of variance with an adjusted *post hoc* test for multiple comparisons as appropriate. Differences were considered statistically significant at *p* values < 0.05. Descriptive statistics were expressed as the mean ± standard error of the mean.

## 3. Results

### 3.1. TSLP induces CCL4 Release under Eosinophilic Airway Inflammation

TSLP mRNA expression significantly increased in eosinophil-rich nasal polyps compared with eosinophil-poor uncinate tissues from the same patients with ECRS (Figure 1A). In addition, peripheral blood eosinophils or EPX enhanced TSLP expression in BEAS-2B cells, suggesting that eosinophilic airway inflammation induced TSLP (Figure 1B,C). TSLP and EPX synergistically enhanced the expression of eosinophilic chemokines such as CCL4, CCL5, CCL11, and CCL26 in BEAS-2B cells, especially CCL4 (Figure 1D,E). Synergistically, EPX increased TSLPR expression in BEAS-2B cells (Figure 1F).

### 3.2. TSLP Enhances TLR3 Ligand-Mediated CCL4 Release

Next, we examined innate immune reactions during CCL4 release from BEAS-2B cells. Among TLR2 ligand zymosan, TLR3 ligand poly I:C, TLR4 ligand LPS, TLR7 ligand imiquimod, and PAR2 agonist 2-Furoyl-LIGRLO-amide, poly I:C significantly increased CCL4 release from BEAS-2B cells (Figure 2A), which was enhanced by TSLP (Figure 2B), possibly via upregulated TLR3 expression which was further strengthened by EPX (Figure 2C). Taken together, CCL4 may be associated with innate immune responses during eosinophilic inflammation. 

### 3.3. CCL4 Exerts Autocrine Effects during Eosinophilic Airway Inflammation

We confirmed the autocrine effects of CCL4 in BEAS-2B cells. CCR5, a specific receptor for CCL4 is expressed on BEAS-2B cells (Figure 3A). CCL4 itself increased CCL4 expression in BEAS-2B cells, which was enhanced by co-incubation with peripheral blood eosinophils (Figure 3B), possibly via eosinophil- or EPX-mediated enhancement of CCR5 expression (Figure 3C,D). Moreover, CCL4 increased ICAM-1 expression in BEAS-2B cells (Figure 3E), which may have facilitated eosinophil localization and activation at the local inflammatory sites.

### 3.4. CCL4 Expression in Nasal Polyps Is Reduced after Dupilumab Treatment

CCL4 was highly, predominantly expressed in nasal polyp epithelial cells from patients with ECRS (upper panels in Figure 4A). Four months after treatment with dupilumab, an IL-4 receptor-α antagonist, CCL4 expression in nasal polyps was significantly reduced (lower panels in Figure 4A). These results were representative of at least four patients who responded to dupilumab (Figure 4B).

### 3.5. Serum CCL4 Correlates with Type 2 Inflammation Markers

Finally, we explored the possibility of serum CCL4 as a biomarker for type 2 airway inflammation. In parallel with FENO and total serum IgE levels, serum CCL4, CCL11, and CCL17 levels, but not CCL26, were significantly reduced 4 months after treatment with dupilumab (Figure 5A–F). Serum CCL4 levels were positively correlated with FENO and total serum IgE levels (Figure 5G,H), whereas other chemokines, such as CCL11, CCL26, and CCL17 were not correlated (Figure 5I). Furthermore, we identified a positive correlation between serum CCL4 and CCL17 levels (Figure 5J), suggesting that serum CCL4 could be a relatively specific marker for type 2 airway inflammation.

## 4. Discussion

TSLP, but not IL-25 and IL-33, was constantly upregulated in eosinophil-rich areas (nasal polyps) of ECRS when compared with eosinophil-poor areas (uncinate tissues of the same patients), and was significantly enhanced by co-incubation with eosinophils or EPX in BEAS-2B cells, suggesting eosinophilic inflammation induced TSLP, consistent with a previous report [15]. Additionally, TLSPR levels were increased in sinonasal mucosa, predominantly in epithelial cells from patients with ECRS, and may provide positive feedback during type 2 inflammation [15]. EPX not only enhanced TSLP release from BEAS-2B cells, but also upregulated the TSLPR in our system, which may account for this positive feedback.

Signaling via TLR3, due to viral infection such as rhinovirus, exacerbates asthma and chronic rhinosinusitis [16,17]. The viral RNA analog poly I:C, TLR3 ligand was reported to induce TSLP in nasal polyp epithelial cells or bronchial epithelial cells in severe eosinophilic asthma [18,19]. Furthermore, poly I:C may lead to type 2 inflammation in synergy with TSLP [20]. We observed that CCL4 was released from bronchial epithelial cells via the synergistic effects of TL3R ligands and TSLP, therefore CCL4 may be also exacerbate eosinophilic airway inflammation.

While dupilumab reduced CCL4 expression in nasal polyps, its mechanism was unclear. As dupilumab exerted oral corticosteroid-sparing effects in patients with severe asthma [21], it may restore steroid sensitivity in type 2 airway inflammation. One possibility is that proinflammatory cytokines and chemokines, including CCL4, are reduced due to the restoration of corticosteroid sensitivity; we recently observed that serum CCL4 levels were significantly reduced in patients with severe asthma and ECRS, but who responded to omalizumab treatment and showed improved corticosteroid sensitivity [12]. In particular, IL-4 inhibition, a key cytokine[ involved in induction of corticosteroid resistance [22], by dupilumab treatment may restore corticosteroid sensitivity.

ICAM-1 is as an adhesion molecule against eosinophils, with adherent eosinophils becoming activated and degranulated, thereby enhancing local allergic inflammation [23,24,25]. CCL4 induces ICAM-1 expression which may facilitate eosinophil adhesion and activation. Activated eosinophils can release CCL4 [11] and eosinophil granule proteins such as EPX. Furthermore, CCL4 is released from airway epithelial cells at local inflammatory sites via stimulation with EPX and CCL4 itself, leading to the accumulation of eosinophils and other type 2 inflammatory cells, including Th2 cells or group 2 innate lymphoid cells (ILC2) which express CCR5 [26].

In eosinophilic airway inflammation, CCL4 is locally released from airway epithelial cells. Then, CCL4 is orchestrally produced in activated eosinophils, lymphocytes, and macrophages which accumulate in airways via CCL4 [11,27]. CCL4 was detectable in serum, the levels of which correlated with FENO, serum total IgE and CCL17 levels. This sequence of events provides evidence why serum CCL4 should be considered as a biomarker for type 2 airway inflammation. Although it was previously reported that patients with severe asthma who were responded to mepolizumab had lower CCL4 levels when compared with non-responders, this observation was largely inconclusive due to a small sample size [28]. However, CCL4 was increased in BALF samples from patients with corticosteroid resistant asthma [29], and treatment with dupilumab is recommended for oral corticosteroid-dependent severe asthma [7,21,30,31], consistent with our findings.

## 5. Conclusions

CCL4 released from epithelial cells may be associated with innate immune responses during eosinophilic airway inflammation. Although our findings suggest the potential of serum CCL4 as a biomarker for type 2 airway inflammation, further studies comparing type 2 with non-type 2 inflammation are required.

## Figures and Tables

**Figure 1 biomedicines-10-01779-f001:**
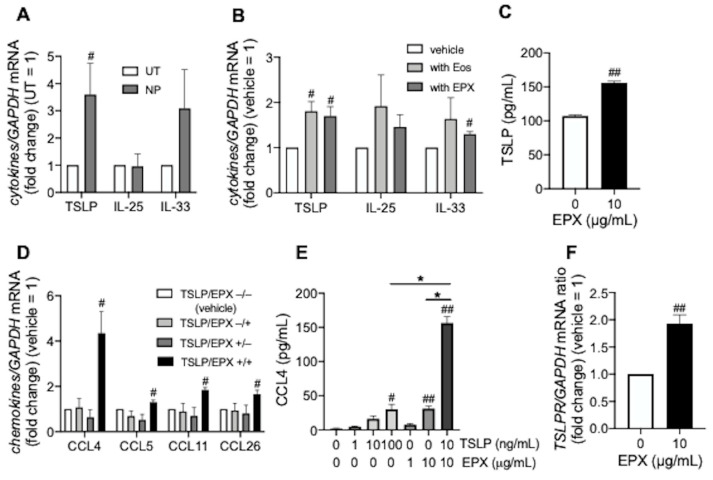
Epithelial cell-derived cytokines and chemokines levels during eosinophilic inflammation. (**A**,**B**) Innate immune response-associated cytokine mRNA in nasal polyps from patients with ECRS (**A**) and BEAS-2B cells co-incubated overnight with purified peripheral blood eosinophils or recombinant eosinophil peroxidase (EPX, 10 µg/mL) for 72 h (**B**). (**C**) TSLP release from BEAS-2B cells incubated with EPX for 48 h. (**D**,**E**) Eosinophil-recruiting chemokines in BEAS-2B cells incubated with TSLP (10 ng/mL) for 48 h, followed by overnight stimulation with EPX (10 µg/mL). CCL4, CCL5, CCL11, and CCL26 mRNA levels (**D**) and CCL4 release (**E**) were evaluated. (**F**) TSLP receptor (TSLPR) mRNA levels in BEAS-2B cells co-incubated with EPX for 48 h. Values in ((**A**,**B**,**D**,**F**) represent ratios to control (uncinate process tissue; UT from the same patients or non-treatment; vehicle) and values in all panels represent the mean ± standard error of the mean from four experiments; ^#^
*p* < 0.05, ^##^
*p* < 0.01 (vs. UT or vehicle), * *p* < 0.05 (as shown between the two groups).

**Figure 2 biomedicines-10-01779-f002:**
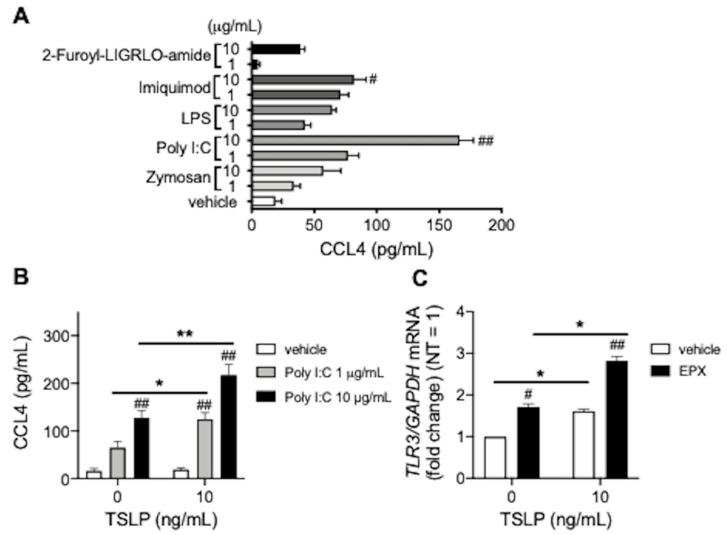
TLR3 ligand-mediated CCL4 release. (**A**) CCL4 release from BEAS-2B cells stimulated overnight with TLR ligands (zymosan, poly I:C, LPS, imiquimod), and the PAR2 agonist (2-Furoyl-LIGRLO-amide). (**B**) CCL4 release from BEAS-2B cells co-incubated overnight with TSLP and poly I:C. (**C**) TLR3 mRNA levels in BEAS-2B cells incubated with TSLP for 48 h, followed by overnight stimulation with EPX (10 µg/mL). Values represent the mean ± standard error of the mean from four (**A**,**B**) or three (**C**) experiments; ^#^
*p* < 0.05, ^##^
*p* < 0.01 (vs. vehicle in each group), * *p* < 0.05, ** *p* < 0.01 (as shown between the two groups).

**Figure 3 biomedicines-10-01779-f003:**
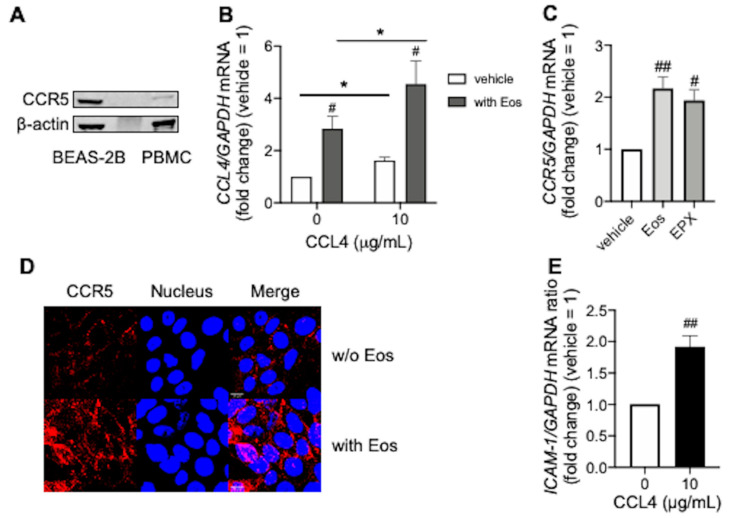
Autocrine effects of CCL4 during eosinophilic inflammation. (**A**) CCR5 expression in BEAS-2B cells. Peripheral blood mononuclear cells (PBMC) were used as a positive control. Experiments were repeated at least three times with different cell preparations. (**B**) CCL4 mRNA levels in BEAS-2B cells co-incubated overnight with CCL4 and/or purified peripheral blood eosinophils. (**C**) CCR5 mRNA levels in BEAS-2B cells co-incubated overnight with purified peripheral blood eosinophils, or EPX (10 µg/mL) for 48 h. (**D**) CCR5 expression in BEAS-2B cells co-incubated overnight with purified peripheral blood eosinophils. CCR5 (red) and the nucleus (blue) are shown in upper (without eosinophils) and in lower (with eosinophils) panels, respectively. Images were obtained using a FV3000 confocal microscope (400× objective). Scale bars in the bottom-left corner of Merge indicate 10 μm. Results were representative of at least three experiments. Values in (**B**,**C**,**E**) represent the mean ± standard error of the mean from four experiments; ^#^
*p* < 0.05, ^##^
*p* < 0.01 (vs. vehicle in each group), * *p* < 0.05 (as shown between the two groups).

**Figure 4 biomedicines-10-01779-f004:**
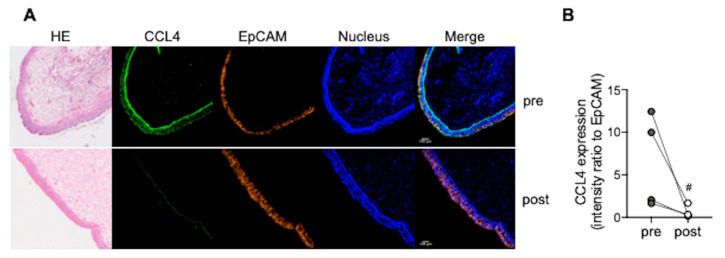
CCL4 expression in nasal polyps. (**A**,**B**) Nasal polyps were obtained from patients with ECRS by excision biopsy under local anesthesia in the outpatient clinic. CCL4 and EpCAM expression in nasal polyps were evaluated pre- and post-treatment with dupilumab. Hematoxylin & Eosin (HE) staining; CCL4 (green), EpCAM (orange), and the nucleus (blue) are shown in upper (pre-treatment) and in lower (post-treatment) panels, respectively (**A**). Images were obtained using a FV3000 confocal microscope (100× objective). Scale bars in the bottom-left corner of Merge indicate 100 μm. CCL4 intensity is expressed as fold change relative to EpCAM (**B**). Individual values are shown; ^#^
*p* < 0.05 (vs. pre-treatment).

**Figure 5 biomedicines-10-01779-f005:**
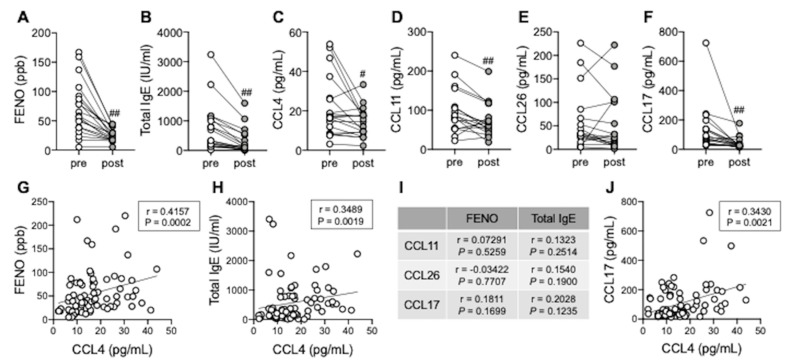
Serum CCL4 levels as a biomarker of type 2 airway inflammation. (**A**–**F**) FENO (**A**), serum total IgE (**B**), CCL4 (**C**), CCL11 (**D**), CCL26 (**E**), and CCL17 (**F**) levels were evaluated pre- and post-treatment with Dupilumab. Individual values are shown (*n* = 19); ^#^
*p* < 0.05, ^##^
*p* < 0.01 (vs. pre-treatment). (**G**,**H**) Correlations between serum CCL4 levels and FENO (**G**) or total serum IgE levels (**H**). (**I**) Correlations between serum CCL11/CCL26/CCL17 levels and FENO or total serum IgE levels. (**J**) Correlations between serum CCL4 and CCL17 levels. Individual values are shown (*n* = 77: samples from patients with severe ECRS and asthma who received any biotherapy).

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
