# Peer review of "CCL4 Functions as a Biomarker of Type 2 Airway Inflammation"

_biomedicines, 2022, doi:10.3390/biomedicines10081779_

Round 1
Reviewer 1 Report
Authors were investigated that roles of CCL4 as biomarker in type 2 airway inflammation. To explain these hypothesis, authors measured CCL4 level and other cytokines from patient samples and cultured cells.. Furthermore, authors described CCL4 release were mediated by various signaling pathway.
So, authors defined that CCL4 was useful biomarker on type 2 airway inflammatory disease cause of closely correlats with FENO, serum IgE, and CCL17 levels in serum
Therefore, this article available to publish with current form to the Journal.
Author Response
We would like to thank for Reviewer’s favorable comments.
Reviewer 2 Report
In this well-written manuscript, the authors focus on CCL4, which is known to be involved in the accumulation and activation of type 2 inflammatory cells, including eosinophils, thus playing a crucial role in bronchial asthma and ERCS. However, the detailed mechanism of CCL4 production still has to be elucidated and its potential as a biomarker of type 2 airway inflammation has not been examined so far. Here, the authors tried to quantify CCL4 mRNA expression and production via the receptors TSLPR, TLRs or PAR2 in a bronchial epithelial cell culture (BEAS-2B) co-incubated with eosinophils or eosinophil peroxidase. Four serum chemokines (CCL4, CCL11, CCL26, and CCL17) and total IgE serum levels were determined, and FENO and CCL4 expression in polyps was measured in patients with asthma and severe ERCS. Based on these experiments, the authors concluded that CCL4 was induced by TSLP under eosinophilic inflammation and that it was released via TLR3 signaling, enhanced by TSLP. Nasal polyp epithelial cells represented the main location of CCL4, and serum levels of this chemokine were reduced after dupilumab treatment. The latter levels were positively correlated with FENO, serum IgE, and CCL17 levels. Thus, in summary, the authors suggest that CCL4 released from epithelial cells via the innate immune system may represent a useful biomarker of type 2 airway inflammation. The experimental design as a combination of in vitro and ex vivo studies is elegant, the data presentation overall is clear. The last remaining points that still should be clarified are highlighted below:
Minor:
- Figure 1: Are there really no error bars in the control group or have they been omitted? While I understand the standardization procedure to make the control equal 1, data presentation would be more transparent if the variability of the control group were also to be included as I assume the mean +- SEM from four experiments also is valid for the controls?
- 2.5 Were the confocal microscopy imaging settings changed between slides or were they identical?
- Typo in line 141: “possibly via upregulated of TLR3 expression“
- Please provide additional clinical data, how much time was between pre- and post-treatment conditions? How were the samples for histology obtained - FESS under general anesthesia in the OR or excision under local anesthesia in the outpatient clinic?
- Another typo in line 225: “a key cytokines involved”
- Table S2: Correct to “Age (years)”, “Body Mass Index (kg/m²)“, “NSAID intolerance“. There are two lines that mention the same numbers regarding “FENO (ppb)”.
